There are amendments to this paper

# A VEGF receptor vaccine demonstrates preliminary efficacy in neurofibromatosis type 2

Ryota Tamura [1], Masato Fujioka[2], Yukina Morimoto[1], Kentaro Ohara[3], Kenzo Kosugi[1], Yumiko Oishi[1], Mizuto Sato[1], Ryo Ueda[1], Hirokazu Fujiwara[4], Tetsuro Hikichi[5], Shinobu Noji[6], Naoki Oishi [2], Kaoru Ogawa[2], Yutaka Kawakami[6], Takayuki Ohira[1], Kazunari Yoshida[1] & Masahiro Toda [1]*

The anti-VEGF antibody bevacizumab has shown efficacy for the treatment of neurofibromatosis type 2 (NF2). Theoretically, vascular endothelial growth factor receptors (VEGFRs)-specific cytotoxic T lymphocytes (CTLs) can kill both tumor vessel cells and tumor cells expressing VEGFRs. Here we show an exploratory clinical study of VEGFRs peptide vaccine in seven patients with progressive NF2-derived schwannomas. Hearing improves in 2/5 assessable patients (40%) as determined by international guidelines, with increases in word recognition scores. Tumor volume reductions of ≥20% are observed in two patients, including one in which bevacizumab had not been effective. There are no severe adverse events related to the vaccine. Both VEGFR1-specific and VEGFR2-specific CTLs are induced in six patients. Surgery is performed after vaccination in two patients, and significant reductions in the expression of VEGFRs in schwannomas are observed. Therefore, this clinical immunotherapy study demonstrates the safety and preliminary efficacy of VEGFRs peptide vaccination in patients with NF2.

[1] Department of Neurosurgery, Keio University School of Medicine, 35 Shinanomachi, Shinjuku-ku, Tokyo 160-8582, Japan. [2] Department of Otorhinolaryngology, Head and Neck Surgery, Keio University School of Medicine, 35 Shinanomachi, Shinjuku-ku, Tokyo 160-8582, Japan. [3] Department of Pathology, Keio University School of Medicine, 35 Shinanomachi, Shinjuku-ku, Tokyo 160-8582, Japan. [4] Department of Radiology, Keio University School of Medicine, 35 Shinanomachi, Shinjuku-ku, Tokyo 160-8582, Japan. [5] OncoTherapy Science, Inc., 3-2-1, Sakado, Takatsu-ku, Kawasaki City, Kanagawa 213-0012, Japan. [6] Division of Cellular Signaling Institute for Advanced Medical Research, Keio University School of Medicine, 35 Shinanomachi, Shinjuku-ku, Tokyo 160-8582, Japan. *email: todam@keio.jp

Neurofibromatosis type 2 (NF2) is associated with the development of schwannomas at multiple sites, including the bilateral vestibular portion and meningiomas[1]. To date, there is no established effective treatment for NF2, because after surgical resection tumors are highly likely to regrow[2]. The use of stereotactic radiosurgery has recently become an effective management modality for NF2 schwannomas[3]. However, radiosurgery is not advocated for multiple or large tumors[3]. Vestibular dysfunction and trigeminal neuropathy have been reported after radiosurgery[4]. While it is evidently uncommon, malignant transformation associated with radiosurgery has also recently been reported[5].

Treatment with the anti-vascular endothelial growth factor (VEGF) antibody bevacizumab has reportedly resulted in tumor control and hearing improvement in NF2 patients[6], presumably because VEGF-A is essential for the growth of these tumors. Some aspects of bevacizumab treatment are problematic, however, such as the need for frequent parenteral administration, side effects, apparent drug resistance, and rebound tumor progression after cessation[7].

The aim of peptide vaccine immunotherapy is to activate cytotoxic T lymphocytes (CTLs) in patients via the administration of antigens in the form of peptides. We recently reported a pilot clinical study investigating VEGF receptor (VEGFR)1 and VEGFR2 peptide vaccination in malignant glioma patients, in which the treatment exhibited safety, and yielded therapeutic effects in some patients[8].

In this study, we demonstrate VEGFRs expression in endothelial cells and tumor cells in NF2 schwannomas, and conduct an exploratory clinical investigation of VEGFRs peptide vaccination in patients with progressive NF2. Differences in VEGF/VEGFR status and key molecules and cells in the tumor microenvironment between NF2 and non-NF2 schwannomas, and between NF2 schwannomas pre-vaccination and post vaccination, are also analyzed. Immunotherapy using VEGFRs peptides is evidently safe and exhibits preliminary efficacy for NF2 patients. Although the efficacy of peptide vaccination alone for malignant diseases has thus far proved to be limited, it may have the capacity to slow the growth of benign tumors, including those in NF2 patients.

## Results

**Patient characteristics.** Seven patients with progressive schwannomas (five of HLA-A*2402 type, one of HLA-A*0206 type, and one of HLA-A*0207 type) were enrolled in this study between September 2016 and March 2018. Three were male and four were female, and their ages ranged from 17 to 41 years. Case 6 had been treated previously with bevacizumab (five times every 2 weeks) at another hospital. All patients received eight VEGFRs peptide vaccinations (Table 1).

**Immunological monitoring.** In six patients (cases 1–4, 6, and 7) CTLs specific for both VEGFR1 and VEGFR2 were induced after the vaccinations, and in one patient (case 5) only CTLs specific for VEGFR1 were induced. In three patients (cases 2–4) analyzed 7 months after the last vaccination, strong CTL responses against VEGFR2 were still detected (Table 1).

**Adverse events.** During the vaccination course, five patients developed grade 1 or grade 2 local skin reactions at the injection sites with induration, redness, and swelling. No patients developed ulceration at the injection sites, and no delayed wound healing or gastrointestinal bleeding was observed. Grade 1 hypertension was noted in one patient. Grade 1 or grade 2 neutropenia and anemia were observed in four patients. Although grade 3 diverticulitis was observed in one patient after four

| Table 1 Patient characteristics. | | | | | | | | | |
|---|---|---|---|---|---|---|---|---|---|
| **Patient**[a] | **PS** | **HLA-A type** | **Peptide** | **Prior therapy** | **ELISPOT (CTL)** | | | **CTL induction** | |
| | | | | | **Mo** | **R1** | **R2** | **R1** | **R2** |
| Case 1 | 3 | 2402/3303 | 2402 | Surgery (bil. ves. sch), Rad (bil. ves. sch) | 0 | 1+ | – | + | + |
| | | | | | 3 | NT | NT | | |
| | | | | | 6 | 3+ | 3+ | | |
| Case 2 | 1 | 2402/3101 | 2402 | Surgery (bil. ves. sch), Rad (Lt. ves. sch) | 0 | – | – | + | + |
| | | | | | 3 | – | 1+ | | |
| | | | | | 6 | 1+ | – | | |
| | | | | | 12 | – | 3+ | | |
| Case 3 | 1 | 2402/0201 | 2402 | Surgery (bil. ves. sch) | 0 | – | – | + | + |
| | | | | | 3 | 3+ | 2+ | | |
| | | | | | 6 | 3+ | 3+ | | |
| | | | | | 12 | – | 2+ | | |
| | | | | | 18 | 3+ | 3+ | | |
| Case 4 | 1 | 2402/3101 | 2402 | Surgery (Rt. ves. sch) | 0 | – | 1+ | + | + |
| | | | | | 3 | NT | NT | | |
| | | | | | 6 | 3+ | 3+ | | |
| | | | | | 12 | 3+ | 3+ | | |
| Case 5 | 1 | 0207/0301 | 0201 | Surgery (FB Men) | 0 | 1+ | 1+ | + | – |
| | | | | | 3 | 2+ | – | | |
| | | | | | 6 | 1+ | 1+ | | |
| Case 6 | 1 | 2402/2420 | 2402 | Surgery (C1), Rad (Lt. ves.sch), Bev[a] | 0 | 2+ | 1+ | + | + |
| | | | | | 3 | 3+ | 2+ | | |
| | | | | | 6 | 1+ | 1+ | | |
| Case 7 | 1 | 0206/2602 | 0201 | Surgery (Rt. ves. sch) | 0 | 2+ | – | + | + |
| | | | | | 3 | – | 3+ | | |
| | | | | | 6 | 3+ | – | | |

*Bev* bevacizumab, *bil* bilateral, *CTL* cytotoxic T lymphocyte, *ELISPOT* enzyme-linked immunospot, *F* female, *FB* frontal base, *Lt* left, *M* male, *Men* meningioma, *Mo* months after the first vaccination, *NT* not tested, *PS* performance status, *Rad* radiosurgery, *R1* vascular endothelial growth factor receptor 1, *R2* vascular endothelial growth factor receptor 2, *Rt* right, *sch* schwannoma, *ves* vestibular
[a]Although four cycles of Bev were administered 3 years before the first vaccination, tumor growth was not regulated

**Table 2 Patient outcome.**

| Patient# | Radiographic response[a] | | Hearing impairment[b] before the trial | Hearing response (WRS) | Hearing response (PTA) | Adverse events |
|---|---|---|---|---|---|---|
| | Schwannoma | Meningioma | | | | |
| Case 1 | Rt. ves: MR<br>Lt. ves: MR<br>Rt. trig: SD<br>Lt. trig: SD | con: MR | Profound, Bil not testable<br>AAO-Class DTokyo-bil: F | NC | Rt.: NC<br>Lt.: NC | Intracerebral hemorrhage (no relationship) |
| Case 2 | Rt. ves: SD<br>Lt. ves: SD<br>Lt. trig: MR | —— | Profound<br>Rt not testable<br>AAO-Class D<br>Tokyo-Rt:F/Lt:D | (+) | Rt.: NC<br>Lt.: NC | No |
| Case 3 | Rt. ves: PD<br>Lt. ves: MR | —— | Profound<br>Bil measurable<br>AAO-Class D<br>Tokyo-Rt: D/Lt:E | NC | Rt.: NC<br>Lt.: NC | No |
| Case 4 | Rt. ves: PR<br>Lt. ves: PR<br>Rt. trig: SD<br>Lt. trig: SD | CP: MR<br>falx: SD | Profound<br>Bil measurable<br>AAO-Class D<br>Tokyo-Rt: D/Lt: F | NC | Rt.: NC<br>Lt.: NC | Diverticulitis (no relationship) |
| Case 5 | Rt. ves: MR<br>Lt. ves: MR | FB: PD | Profound<br>Bil measurable<br>AAO-Class D<br>Tokyo-bil: D | excluded | Rt.: NC<br>Lt.: NC | No |
| Case 6 | Rt. ves: MR<br>Lt. ves: MR<br>Rt. trig: PR<br>Lt. trig: PR | —— | Lt good<br>Rt. Profound<br>AAO-Class Rt: D/Lt: A<br>Tokyo-Rt: F/Lt: A | excluded | Rt.: (−)<br>Lt.: NC | No |
| Case 7 | Rt. ves: MR<br>Lt. ves: MR<br>Rt. trig: SD<br>Lt. trig: MR | —— | Profound, Rt not testable<br>AAO-Class D<br>Tokyo-Rt: F/Lt: D | (+) | Rt.: NC<br>Lt.: NC | No |

*AAO* American Academy of Otolaryngology–Head and Neck Surgery classification, *CP* cerebellopontine angle, *FB* frontal base, *Lt* left, *NC* no change, *PTA* pure-tone audiogram, *Rt* right, *Tokyo* classification, *trig* trigeminal, *ves* vestibular, *WRS* word recognition score
[a]Radiographically determined response was evaluated based on the parameters described in a previously reported clinical trial: partial response (PR) = decrease in tumor volume of 20% or more; minor response (MR) = decrease in tumor volume of 5% to 19%; progressive disease (PD) = increase in tumor volume of 20% or more; and stable disease (SD) if none of the aforementioned applied[9]
[b]Hearing abilities in neurofibromatosis type 2 patients were classified using the American Academy of Otolaryngology–Head and Neck Surgery (AAO-HNS) classification system (class A–D) and Tokyo classification system (class A–F). Improvement or deterioration in hearing was defined as a change in WRS as defined in a previous study, and is indicated as (+/−)[15]. In the evaluation of pure-tone audiograms, improvement or deterioration of hearing was defined as a change of at least 10 dB from 1 to 3 kHz. Deterioration in hearing is indicated by (−)

vaccinations, this adverse event was presumably not related to vaccination because diverticulitis had previously occurred every few months in this patient. Grade 5 intracranial hemorrhaging was observed in one patient 5 months after the last vaccination. Subarachnoid hemorrhage around the left trigeminal schwannoma and the surface of the left temporal lobe was observed on CT and MRI. It has been reported that intratumoral hemorrhage and subarachnoid hemorrhage occur in 1%–8% of schwannoma patients[9]. We performed surgery to reduce the mass effect and pressure in the brain, but intratumoral hemorrhage was not detected during the operation. Detailed image analysis revealed dural arteriovenous fistula (dAVF) around the left temporal lobe at the time of hemorrhage, and also prior to vaccination. dAVF may have been the origin of hemorrhage. The aforementioned Independent Data Monitoring Committee concluded that the experimental vaccine was not likely to be related to this adverse event (Table 2).

**Clinical response.** Based on previously defined criteria[10] two cases (cases 5 and 6) who exhibited WRSs ≥90 pre-vaccination were excluded from hearing evaluation in this study. Improvements in WRS were observed in two of the five assessable cases (40%), and there was no change in the other three cases. Ordinarily, the natural course of hearing loss in NF2 patients is progressive deafness (Fig. 1f, g, Table 2). Average PTA thresholds at 1, 2, and 3 kHz were calculated in 14 assessable ears (seven cases) before vaccination and after all 8 vaccinations. The threshold

deteriorated in one ear (the right ear of case 6, in whom bevacizumab had not been effective) in a patient who had already exhibited 0% in WRS testing prior to vaccination. In contrast, case 6's left ear exhibited improvement in average PTA thresholds (Fig. 1h). The patient and her family noted hearing improvement in her left ear 4 weeks after the first vaccination.

Of the total of seven patients, PR was observed in more than one schwannoma in two (case 6 in whom previous bevacizumab treatment had been ineffective, and case 4), and MR were observed in the other five (Fig. 1a, Table 2; Supplementary Fig. 1). Out of 23 schwannomas, PR were observed in 4, MR in 11, and 7 exhibited SD (Fig. 1b, c). Only one cystic tumor became enlarged 12 months after the first vaccination (PD) (see Supplementary Fig. 1). Then, the growth of cystic tumor had stopped. Percent change in tumor volume significantly decreased after vaccination in all seven patients (mean 1 year before vaccination 39.75 ± 27.74%, mean 1 year after vaccination −4.67 ± 15.86, $p = 0.0087$) (Fig. 1d). In TBV analysis, reductions of >30% were observed in five tumors, reductions of >10% but <30% were observed in five tumors, and increases of >10% were observed in three tumors (Fig. 1a, e).

With regard to overall meningioma size, MR were observed in two, SD was observed in one, and PD was observed in one (Fig. 2a, b, c, Table 2; Supplementary Fig. 1). Percent change in tumor volume reduced after vaccination in three patients (mean 1 year before vaccination 213.25 ± 173.8%, mean 1 year after vaccination 20.71 ± 21.32%, $p = 0.064$; Fig. 2d). In TBV analysis, one tumor exhibited a >30% decrease. However, other three

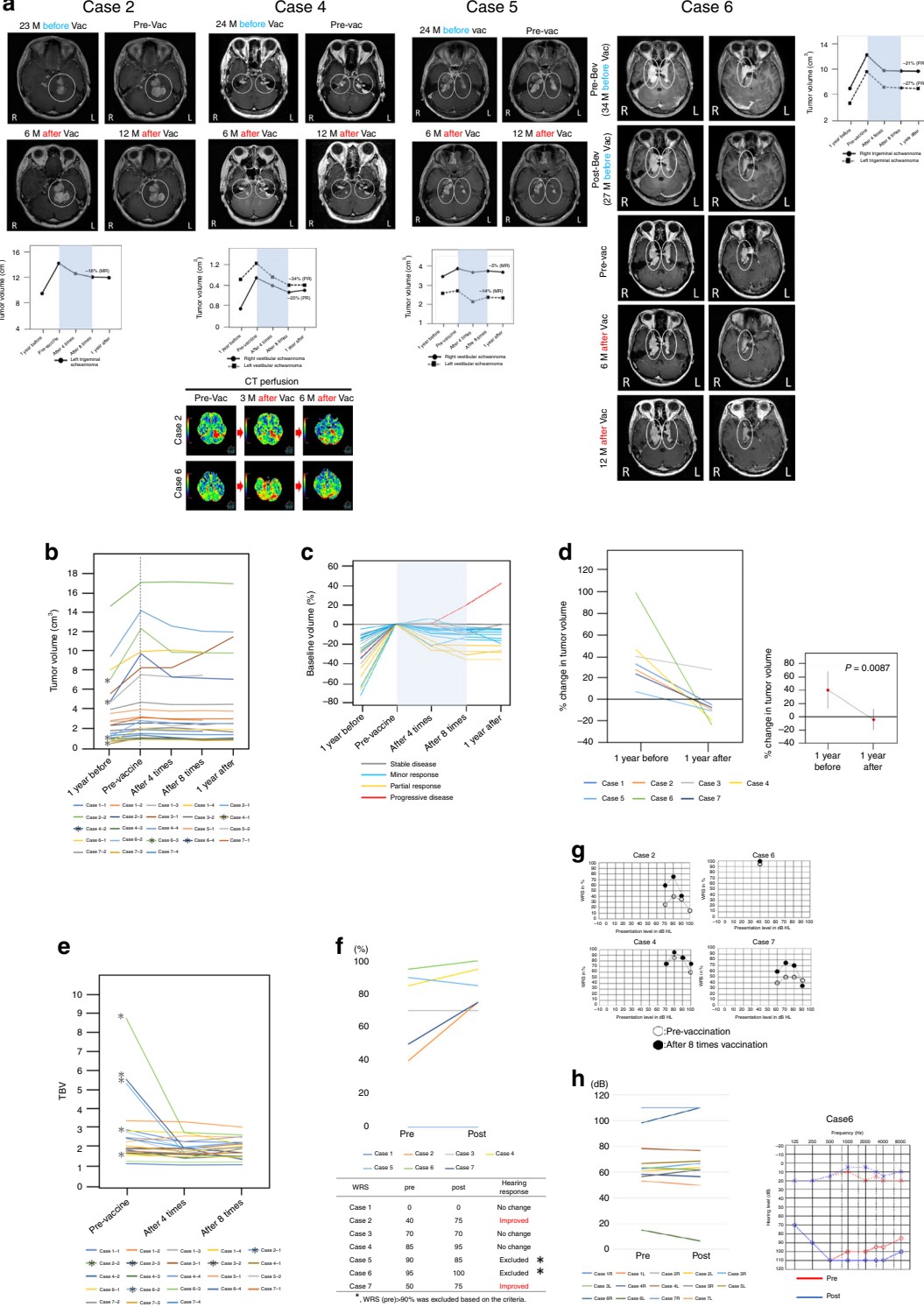

tumors did not keep decreasing during the course of eight vaccinations (Fig. 2e).

**Histopathological analysis of schwannomas**. Most vessels in schwannomas in non-NF2 patients exhibited slight or negative VEGFRs expression (Fig. 3a). These vessels were covered with pericytes expressing PDGFR-β. In contrast, many vessels in schwannomas in NF2 patients exhibited strong VEGFRs expression without PDGFR-β-positive pericytes (Fig. 3a). More importantly, VEGFR1 and VEGFR2 were expressed on tumor cells in schwannomas in NF2 patients (Fig. 3c), suggesting that NF2 schwannomas themselves can be direct targets of VEGFRs vaccination. There was no significant difference in MVD between schwannomas in non-NF2 patients and NF2 patients (Fig. 3c, d). However, the diameter of vessels was significantly larger in NF2 patients than in non-NF2 patients (Fig. 3c, d), and VEGF-A expression was stronger in NF2 patients (Fig. 3c). Although the differences were not statistically significant, the relative gene expression of VEGF-A and VEGFRs tended to be higher in NF2

**Fig. 1 Radiographic analysis of schwannomas after vaccination. a** Gd-enhanced magnetic resonance imaging (MRI) and computed tomography (CT) perfusion scans in cases 2, 4, 5, and 6. White circles indicate tumors targeted for volumetric analysis. CT perfusion revealed reduced tumor blood volume (TBV) in case 2, and 6 after the first vaccination. The range of the color scale is shown. **b** All 23 schwannomas exhibited increased volume during the year preceding the first vaccination. Volumetric schwannoma reductions of ≥20% (PR) were observed in four tumors (*). **c** Percent change in tumor volume from baseline (pre-vaccination) in all schwannomas. The median baseline target schwannoma volume was 2.73 cm$^3$ (range 0.78–14.08 cm$^3$). **d** Percent change in tumor volume in seven patients, and mean values 1 year before and 1 year after the first vaccination. Percent change in tumor volume decreased significantly after vaccination in all schwannomas in seven patients. P-value was determined by paired t test. **e** TBV changes in all 23 schwannomas. Reductions of >30% were observed in five tumors (*). **f** Hearing improvement of word recognition scores (WRSs) was observed in two of the five assessable cases (40%). **g** Changes in WRSs after eight vaccinations in cases 2, 4, 6, and 7. These patients were profoundly deaf in one ear but not the other, and their respective improvements in WRSs in their hearing ears were from 40% to 75%, 85% to 95%, 95% to 100%, and 50% to 75%. **h** Hearing changes in the average of PTA. Thresholds in PTA did not change in 13 of 14 ears. The threshold deteriorated in 1 ear (right ear of case 6), which had already been 0% in WRS before vaccination. In contrast, left ear of case 6 showed some improvement of the average of PTA. Bev bevacizumab, GD gadolinium, L left ear, MRI magnetic resonance imaging, M months, CT computed tomography, R right ear, TBV tumor blood volume, Vac vaccine, Pre-Vac the day of the first vaccination, 1 year after 1 year after the first vaccination, 1 year before 1 year before the first vaccination.

patients than in non-NF2 patients, which is consistent with the immunohistochemistry results (Fig. 3b, c). In a previous study, the mean growth rate of schwannomas in NF2 patients was 2 mm/year[11]. Therefore, we deemed growth of >2 mm/year indicative of a progressive course. In NF2 patients, the number of Foxp3-positive cells in schwannomas with a progressive course was significantly higher than in those without a progressive course, suggesting that growth may be associated with Foxp3-positive regulatory T cells (Tregs) (see Supplementary Fig. 2a). Low PD-L1 expression was observed on tumor cells or endothelial cells in NF2 patients. The PD-L1 scores were 0 in 14/23 tumors and 1 in 9/23 tumors (see Supplementary Fig. 2b).

Histological changes in schwannomas post vaccination could be analyzed in cases 1 and 2. In both cases, most tumor vessels exhibited strong VEGFRs expression without PDGFR-β-positive pericytes before vaccination (Fig. 4a, c). In contrast, after vaccination tumor vessels exhibited negative or slight VEGFRs expression, and most endothelial cells were covered with PDGFR-β-positive pericytes (Fig. 4a, c). VEGF-A expression was also lower after vaccination than it was before vaccination (Fig. 4e). In both cases, in qPCR analysis the mean relative gene expression levels of *VEGF-A* and *VEGFRs* in tumors were lower after vaccination than before vaccination (Fig. 4b, d). Vessel diameter was smaller after vaccination, and MVD was lower (Fig. 4f). The numbers of Foxp3-positive cells and *FOXP3* gene expression were lower in tumors after vaccination (Fig. 4g). Although the total numbers of CD8-positive cells did not change substantially after vaccination, the numbers of CD8-positive cells in the perivascular area were increased (Fig. 4h). In immunofluorescence analysis, many CD8-positive cells were observed around vessels with weak VEGFRs expression after vaccination (see Supplementary Fig. 3). After vaccination, greater numbers of diffuse cleaved caspase 3-positive cells were detected in tumors that included endothelial cells (Fig. 4i), suggesting that vaccination induced apoptosis in tumors and tumor vessels.

## Discussion

Because schwannoma growth mainly depends on the VEGF-A/VEGFR pathway[12], bevacizumab has demonstrated therapeutic efficacy in NF2 patients. A review of the literature suggests that schwannoma shrinkage and hearing improvement occur in >50% of progressive NF2 patients treated with bevacizumab (see Supplementary Table 2)[6,13–20]. There are some problems associated with bevacizumab treatment, however, such as the need for frequent parenteral administration and side effects. Because tumor growth frequently recurs following the discontinuation of bevacizumab, it is difficult to decide if and when to discontinue bevacizumab treatment in NF2 patients[7].

Clinical trials of peptide-based vaccine therapy using VEGFR-derived epitopes have previously been conducted to assess safety, tolerability, and potential clinical activity in patients with advanced pancreas, gastrointestinal, and renal cell cancers (see Supplementary Table 3)[21–28]. In addition, we previously conducted exploratory clinical trials investigating VEGFRs peptide vaccination with and without multiple glioma oncoantigens in patients with recurrent high-grade gliomas, wherein treatment exhibited safety and yielded therapeutic effects in some patients[8,29]. In one clinical trial using only VEGFRs peptide vaccination in patients with recurrent high-grade glioma, eight patients received weekly vaccinations for 8 weeks. The first four vaccines induced positive immune responses against at least one of the targeted VEGFR epitopes in 87.5% of patients. The median overall survival time in all patients was 15.9 months, which was prolonged compared with historical data. Two patients remained progression-free for at least 6 months. During the vaccination period, one patient developed a grade 3 ulcer at the injection site[8], but no other adverse events were observed during that study. Based on those results, a clinical trial of VEGFRs peptide vaccination is now underway in patients with primary glioblastoma.

In this study, VEGFRs peptide vaccine was administered to NF2 patients. CTL induction was identified in all patients, and no severe adverse events associated with the vaccine were observed. Moreover, two of the five assessable cases exhibited improvements in WRSs, and these improvements evidently had a substantial effect on quality of life. No substantial changes in WRS were observed in the other three cases, including case 1. It may be that there was little chance of any intervention improving hearing in case 1, because he had been profoundly deaf in both ears for a long time (so-called dead ears). It is well known that long-term profound deafness leads to the reduction of cortical hearing perception, and that often such changes cannot be reversed even if there is recovery of input from the auditory nerve. Volumetric radiographic responses (PR) were seen in 2/7 patients, and percent change in tumor volume of all schwannomas and meningiomas decreased after vaccination, indicating acceptable treatment efficacy equivalent to that observed in previous studies involving bevacizumab treatment in NF2 patients (see Supplementary Fig. 2). Notably, of 23 schwannomas, only one cystic schwannoma became enlarged. Cystic schwannomas tend to become enlarged rapidly[30], and therefore the above-described immunotherapy may not be appropriate in cases involving rapidly growing schwannomas with cystic change.

CTLs induced by the vaccine can directly kill a wide variety of cells associated with tumor growth, including tumor vessels, tumor cells, and Tregs. In this study, high VEGFR1 and VEGFR2 expression was detected not only on endothelial cells but also

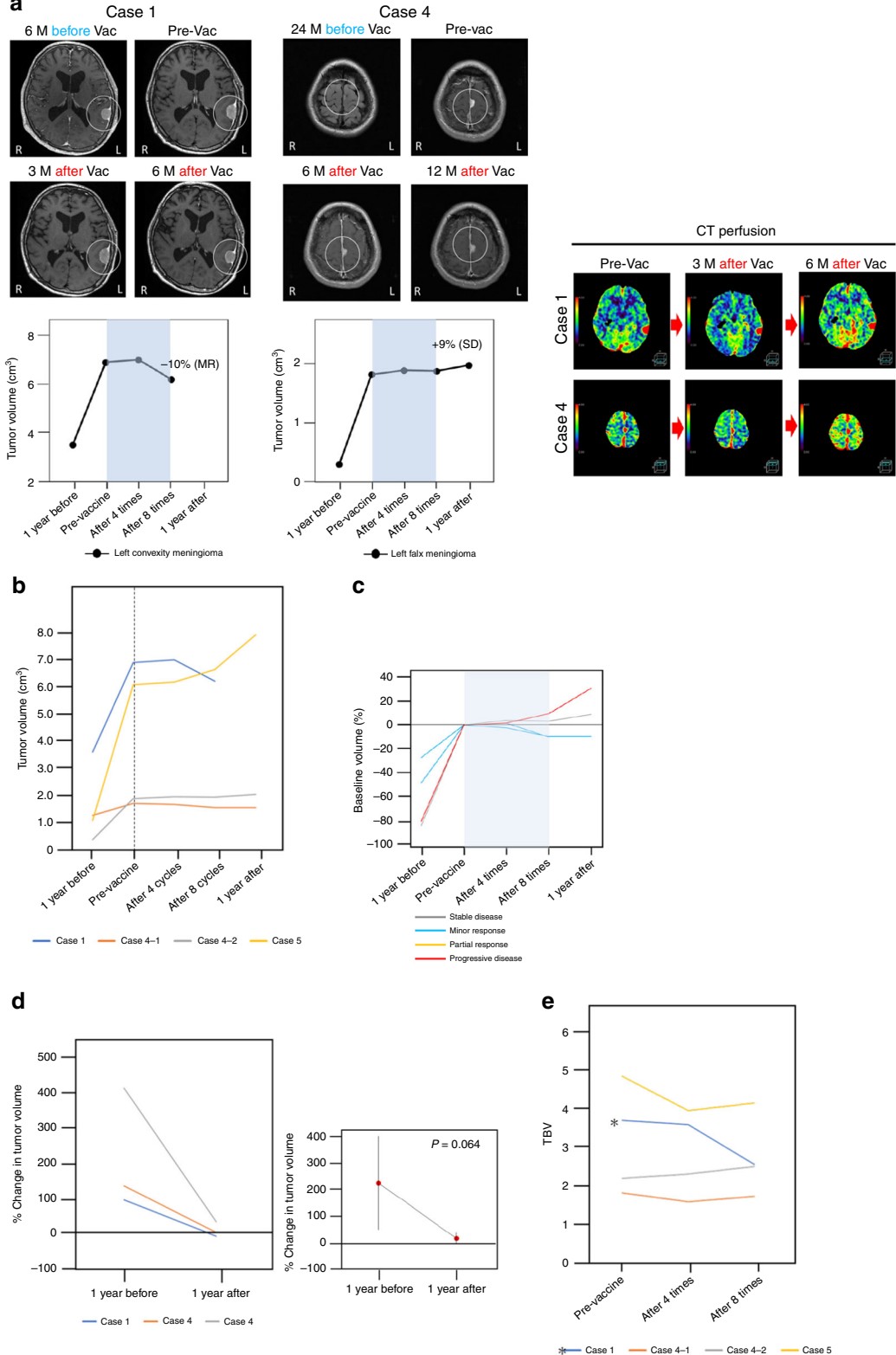

tumor cells in NF2 schwannomas, which is consistent with previous reports[18]. Tregs are known to express VEGFR2[31]. The histological changes observed in tumors after vaccination in this study—specifically, reduced VEGFRs expression, reduced MVD, and normalization of vascular structure with pericyte recovery in schwannomas—are concordant with antitumor mechanisms induced by a VEGFRs-based vaccine. More apoptosis was detected in schwannomas post vaccination, suggesting a

CTL-mediated killing effect. The number of Foxp3-positive cells and *FOXP3* gene expression decreased after vaccination, suggesting that the induced CTLs also killed VEGFR2-expressing Tregs. The paired pre-vaccination/post-vaccination tissues from the same patients were valuable due to the relative rarity of such potentially informative samples, and they were powerful tools with regard to investigation of the biological effects of targeted therapies. In case 6 a better treatment outcome resulted from

**Fig. 2 Radiographic analysis of meningiomas after vaccination. a** Images of Gd-enhanced magnetic resonance imaging (MRI) and computed tomography (CT) perfusion in cases 1 and 4. Although the images 24 months before vaccination are T2-weighted images in case 4, the tumors can be clearly seen. White circles indicate targeted tumors for volumetric analysis. CT perfusion revealed reduced tumor blood volume (TBV) in the left meningioma in case 1 and gradually increasing TBV in the falx meningioma in case 4. The range of the color scale is shown. **b** Changes in tumor volume in the four meningiomas. All meningiomas exhibited increased volume before vaccination. Volumetric meningioma reduction of 10% (MR) was observed in two tumors, a 9% increase (classified as SD in accordance with above-described criteria) was observed in one tumor, and a 31% increase (PD) was observed in one tumor. **c** Percent change in tumor volume from baseline (pre-vaccination) in all meningiomas. The median baseline target meningioma volume was 3.96 $cm^3$ (range 1.65–6.93 $cm^3$). **d** Percent change in tumor volume in three patients and the mean values 1 year before and 1 year after the first vaccination. Percent change in tumor volume decreased after vaccination in all meningiomas. $P$-value was determined by paired $t$ test. **e** TBV changes in the four meningiomas. The left meningioma in case 1 exhibited a decrease of >30%. However, the other three tumors did not keep decreasing during the course of eight vaccinations. GD gadolinium, MRI magnetic resonance imaging, M months, CT computed tomography, TBV tumor blood volume, Vac vaccine, Pre-Vac the day of the first vaccination, 1 year after 1 year after the first vaccination, 1 year before 1 year before the first vaccination.

VEGFRs peptide vaccine than from bevacizumab treatment, suggesting that VEGFRs vaccination can target a larger variety of cells than bevacizumab.

Peptides containing only CD8-positive T-cell epitopes have proved relatively ineffectual against malignant tumors in some previous studies[22–28], and thus to date no such peptide-based vaccines have been approved by the Food and Drug Administration of the USA. Although CD8-positive CTLs were the main focus of antitumor vaccination in this study, vaccines targeting CD4-positive T-helper cells can also generate antitumor responses. CD4-positive T-helper cells can exhibit direct antitumor effects in addition to their helper function[32,33]. Dual activation of tumor-associated antigen-specific CTLs and T-helper cells by dendritic cell vaccines has induced clinical responses that were superior to those induced by a single CTL epitope vaccine in patients with cancers such as melanoma[32–38]. These observations suggest that a combination of CD8 and CD4 epitopes would increase the efficacy of peptide vaccination in NF2 patients. PD-L1 suppresses antitumor immunity and promotes tumor progression because it tolerizes tumor-reactive T cells[39]. In this study, low PD-L1 expression was observed in the schwannomas of most NF2 patients. In a previous study, PD-1 or PD-L1 blockade therapy was a preferable treatment option, even in patients who were PD-L1 negative[40]. Immune checkpoint inhibition may exert synergic effects when administered with this type of CTL-mediated antitumor immunotherapy in NF2 patients.

The observation that hearing improved in two cases (cases 2 and 7) without remarkable tumor volume reduction is suggestive. As well as direct compression of auditory nerve fibers by tumors, in cases of NF2-associated deafness detrimental paracrine substances such as proinflammatory cytokines from tumors have been proposed as a mechanism of cochlear hearing loss[41]. During vaccination in this study, PTA thresholds did not change in 13 of 14 ears (the threshold deteriorated in one ear). In contrast, WRS either improved or remained stable. These observations imply that the ameliorating effects of VEGFR peptides may be due to a reduction in paracrine effects associated with tumors. In a previous study, VEGF/VEGFR signaling was associated with proinflammatory cytokines[42]. Further investigation is warranted to confirm the findings of this study.

The effects of vaccination on meningiomas are noteworthy. In the previous study, volumetric radiographic responses were only seen in 29% of the meningiomas in NF2 patients[43]. That result suggests that activation of the VEGF pathway is not the main driver of angiogenesis in meningiomas[43]. In this study, volumetric radiographic responses were limited in meningiomas.

With regard to potential advantages of VEGFRs peptide vaccination, continuous administration may not be necessary because memory CTLs persist in the long-term. In this study, CTLs specific for VEGFR2 were detected in three patients 7 months after their final vaccination, and the inhibition of tumor growth was maintained.

The main limitation of this study was the small sample size. The prevalence of NF2 is reportedly approximately one in 25,000 to one in 33,000[44]. In addition, HLA-type matching restricted the number of patients enrolled. Although studies with a larger number of NF2 patients are warranted to confirm the findings of this study, the results suggest that this immunotherapeutic strategy may be favorable in patients with benign tumors such as NF2.

## Methods

**Trial oversight**. This was an exploratory clinical study of VEGFRs peptide vaccination for progressive NF2. All protocols were approved by the Keio University Ethics Committee, and conducted in accordance with the Helsinki declaration on experimentation on human subjects. The trial was registered at UMIN (UMIN000023565). Adherence to the trial protocol and accuracy of the completed case report forms and the electronic data sets were assessed at a minimum of three external monitoring visits. The Keio University Clinical and Translational Center, which acts as an independent academic contract research organization, performed the monitoring.

The authors affirm that human research participants provided informed consent to participate in the study and for publication of their data. This is an intermediate reporting, which was authorized by the Keio University Ethics Committee.

**Patients**. NF2 patients diagnosed with progressive schwannoma were enrolled in this clinical study at the Department of Neurosurgery, Keio University School of Medicine. Inclusion criteria included positive genomic DNA typing for HLA-A*2402, 0201, 0206, and 0207 (HLA Laboratory, Kyoto, Japan). It is known that NF2-related tumor sometimes exhibits prolonged stability without any intervention[45]. Therefore, progressive growth was incorporated as an inclusion criterion in this study. Additional inclusion and exclusion criteria details are provided in the Supplementary Table 1.

**Peptides**. Good manufacturing practice (GMP)-grade VEGFR1-A24-1084 peptide (SYGVLLWEIF) and VEGFR2-A24-169 peptide (RFVPDGNRI) were synthesized by BCN Peptides S.A. (Barcelona, Spain), and GMP-grade VEGFR1-A02-770 peptide (TLFWLLLTL) and VEGFR2-A02-773-2L peptide (VLAMFFWLL) were synthesized by PolyPeptide Laboratories (San Diego, USA). HLA-A*0201-restricted peptide was administered to patients with HLA-A*0201, 0206, and 0207, because HLA-A*0201-restricted peptide has been shown to bind to these HLA-A subtypes[46]. Peptides (2 mg) were injected subcutaneously at infra-axillary and inguinal lymph nodes four times every week, then four times monthly thereafter (a total of eight times; see Supplementary Methods).

**Outcomes and assessments**. Primary outcome was the safety of the vaccine. Secondary outcomes were clinical efficacy parameters, including tumor size, hearing ability, and immunological response. Toxicity was assessed using the Common Terminology Criteria for Adverse Events version 4.0 at each visit. To evaluate clinical responses, computed tomography (CT), magnetic resonance imaging (MRI), and hearing examinations were performed within 2 weeks before the first vaccination, after five vaccinations (at the 3-month timepoint), after eight vaccinations (at the 6-month timepoint), and 12 months after the first vaccination. A volume segmentation method based on gadolinium-enhanced T1-weighted images was performed to obtain complex shape data and identify slow growth of NF2-related tumors (see Supplementary Methods and Supplementary Fig. 4)[1,45,47,48]. Percent change in tumor volume 1 year before and after vaccination was measured[49]. In this study, radiographic responses were evaluated in the same way as they were in a previous clinical trial, wherein partial response (PR) was defined as tumor volume reduction of 20% or more, minor response (MR) as

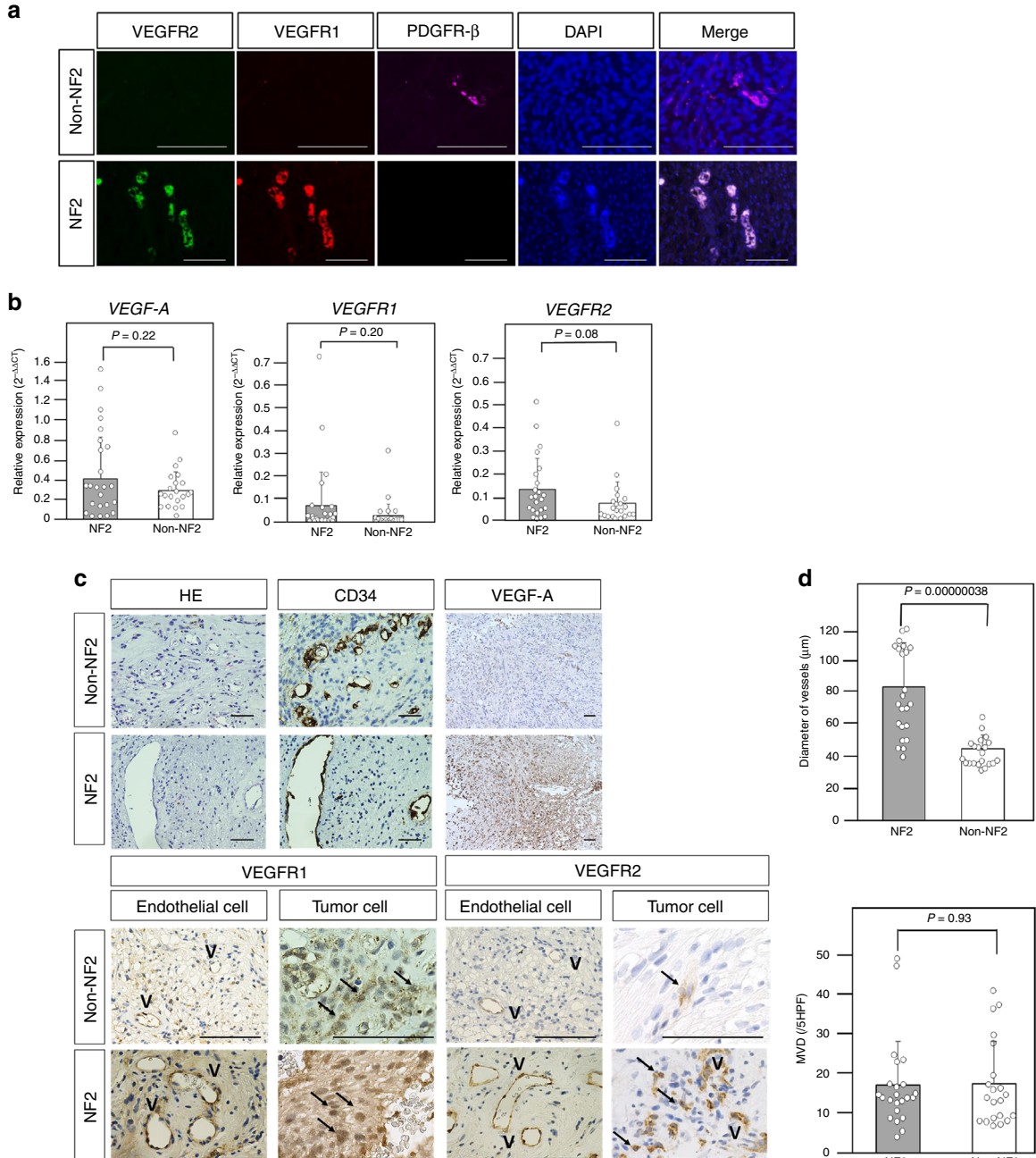

**Fig. 3 Schwannomas in neurofibromatosis type 2 (NF2) and non-NF2 patients. a** Immunofluorescence analysis of vascular endothelial growth factor receptor (VEGFR) 1, VEGFR2, and platelet-derived growth factor receptor-beta (PDGFR-β) expression in tumor vessels in NF2 and non-NF2. Vessels in schwannomas of non-NF2 exhibited negative VEGFR expression with pericytes expressing PDGFR-β. In contrast, vessels in schwannomas of NF2 exhibited strong VEGFR expression without PDGFR-β-positive pericytes. The original magnification was × 400, and the magnification bars = 100 μm. **b** Quantitative real-time PCR analysis of vascular endothelial growth factor (*VEGF*) *A*, *VEGFR1*, and *VEGFR2*. The relative gene expression of *VEGF-A* and *VEGFRs* tended to be higher in NF2 than in non-NF2. *P*-values were determined by Student's *t* test. The mean (bar) ± SD (error bars) is shown (NF2, *n* = 23; Non-NF2, *n* = 21). **c** Hematoxylin and eosin (H&E) staining, and immunohistochemical analysis. The V symbols indicate vascular structure, and the black arrows indicate tumor cells with positive VEGFR1 and VEGFR2 expression. Larger vessels were observed in NF2 than in non-NF2. VEGF-A expression was stronger in NF2. VEGFR1 and VEGFR2 were expressed on endothelial cells and tumor cells in schwannomas in both NF2 and non-NF2. In the panels showing H&E, CD34, VEGFR1, and VEGFR2, the original magnification was ×400 and the magnification bars = 100 μm. In the panels showing VEGF-A, the original magnification was ×200 and the magnification bars = 100 μm. **d** Comparisons of vessel diameter and microvessel density in NF2 and non-NF2. There was no significant difference in microvessel density between schwannomas in non-NF2 and NF2. However, vessel diameter was significantly larger in NF2 than in non-NF2. *P*-values were determined by Student's *t* test. The mean (bar) ± SD (error bars) is shown (NF2, *n* = 23; Non-NF2, *n* = 21). VEGFR vascular endothelial growth factor receptor, PDGFR-β platelet-derived growth factor receptor-beta, DAPI, 4′,6-diamidino-2-phenylindole, NF2 neurofibromatosis type 2, VEGF vascular endothelial growth factor, H&E hematoxylin and eosin, MVD microvessel density, HPF high-power field.

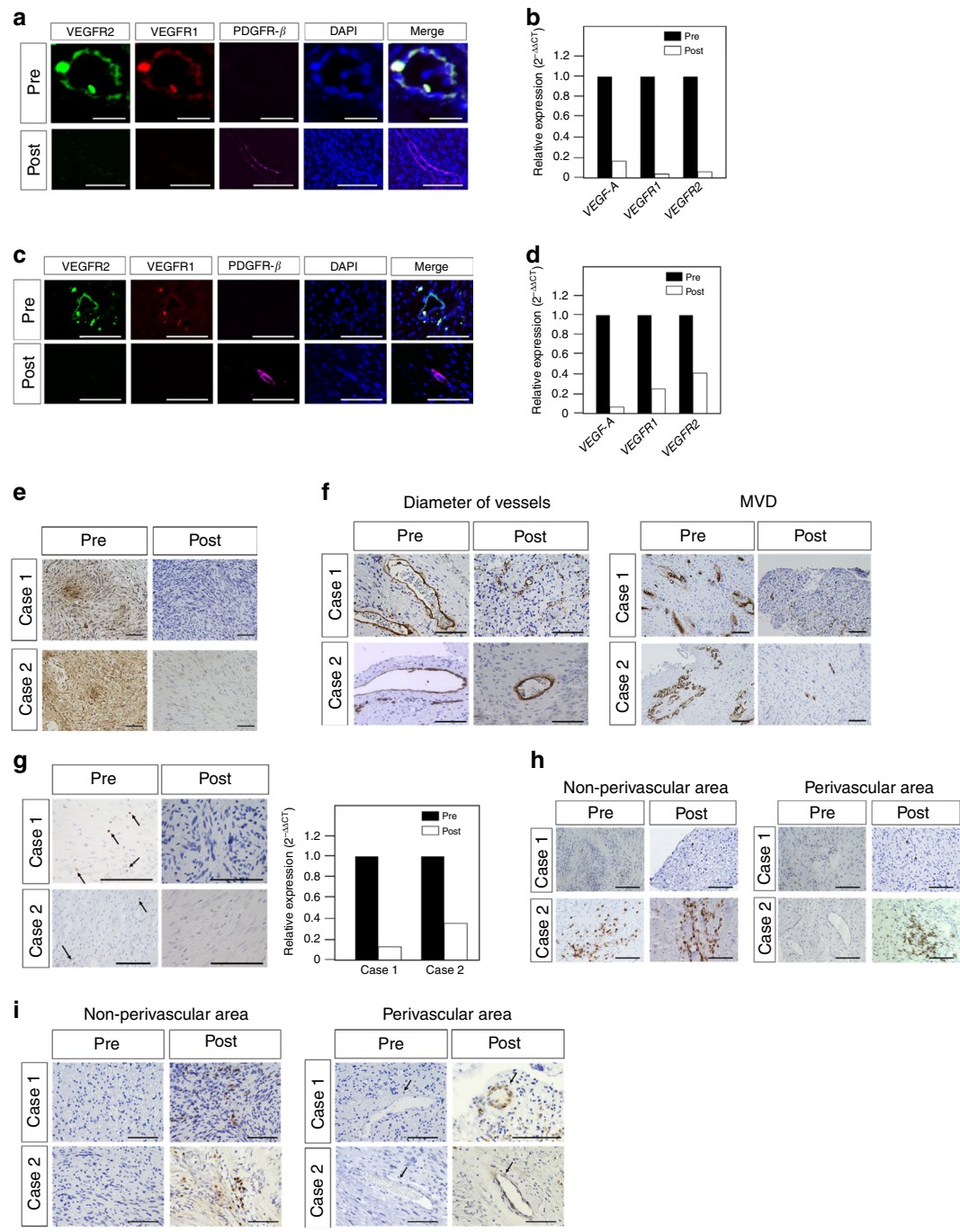

**Fig. 4 Changes in neurofibromatosis type 2 schwannomas after vaccination. a, c** Immunofluorescence analysis revealed decreased expression of vascular endothelial growth factor receptor (VEGFR) 1, VEGFR2, and increased expression of platelet-derived growth factor receptor-beta (PDGFR-β) in schwannomas post vaccination in case 1 (**a**) and case 2 (**c**). **b, d** Quantitative real-time PCR analysis showed decreased mean relative expression of vascular endothelial growth factor *(VEGF)-A*, *VEGFR1*, and *VEGFR2* genes in schwannomas post vaccination in case 1 (**b**) and case 2 (**d**) (pre, $n = 1$ for cases 1 and 2; post, $n = 1$ for cases 1 and 2, respectively). **e** VEGF-A expression was lower after vaccination than before vaccination in cases 1 and 2. **f** Vessel diameter was smaller, and microvessel density was lower after vaccination in cases 1 and 2. **g** The numbers of Foxp3-positive cells and *FOXP3* gene expression were lower in tumors after vaccination in cases 1 and 2 (pre, $n = 1$ for cases 1 and 2; post, $n = 1$ for cases 1 and 2, respectively). The black arrows indicate Foxp3-positive cells. **h** The total numbers of CD8-positive cells did not change substantially after vaccination. The numbers of CD8-positive cells in the perivascular area increased in case 2. **i** After vaccination, more cleaved caspase 3-positive cells were detected in tumor cells and endothelial cells in cases 1 and 2. In the panels showing VEGF-A and microvessel density, the original magnification was ×200 and the magnification bars = 100 μm. In all other panels, the original magnification was ×400 and the magnification bars = 100 μm. VEGFR vascular endothelial growth factor receptor, PDGFR-β platelet-derived growth factor receptor-beta, DAPI 4′,6-diamidino-2-phenylindole, VEGF vascular endothelial growth factor, MVD microvessel density.

tumor volume reduction of 5–19%, progressive disease (PD) as tumor volume increase of 20% or more, and stable disease (SD) was recorded if none of the above applied[45,48]. Radiographic responses were confirmed at the end of the study. Tumor blood volume (TBV) was measured via CT perfusion[50]. Hearing responses were evaluated via maximum word recognition scores (WRSs)[10,48]. Improvement or deterioration in hearing were defined as previously described[10,48]. Standard language differences were considered during WRS testing[51]. Pure-tone audiogram (PTA) testing was also performed in this study, because PTAs have been used widely throughout the world with similar sound input (see Supplementary Methods). Peptide-specific immunological responses were analyzed via the enzyme-linked immunospot assay (see Supplementary Methods).

**Laboratory studies.** To investigate the angiogenic pathway and the tumor immune microenvironment with regard to NF2 schwannomas, standard immunohistochemistry was performed for 25 tumors derived from 11 NF2 patients and 21 tumors derived from 21 non-NF2 patients. Four of the 25 tumor samples were paired pre- and post-vaccination samples obtained from two NF2 patients (see Supplementary Methods). Both these tumors exhibited minimal volume change (not progressive disease). Expression levels of VEGF-A, VEGFR1, VEGFR2, CD34, platelet-derived growth factor receptor-beta (PDGFR β), CD8, Foxp3, programmed cell death ligand 1 (PD-L1), and cleaved caspase 3 were examined via immunohistochemistry. Concurrent immunofluorescence staining for VEGFR1, VEGFR2, and PDGFR-β expression and VEGFR1, VEGFR2, and CD8 expression was performed to evaluate vascular characteristics. Quantitative real-time PCR (qPCR) was performed to analyze the gene expressions of *VEGF-A*, *VEGFR1*, *VEGFR2*, and *FOXP3* (see Supplementary Methods).

**Statistics and reproducibility.** Student's *t* test was used to compare Foxp3-positive cell counts, microvessel density (MVD), vessel diameter, and the expression of *VEGF-A*, *VEGFR1*, and *VEGFR2* analyzed via qPCR in NF2 and non-NF2 patients. The paired *t* test was used to assess radiographic changes post vaccination. In this study, treatment response was statistically assessed by comparing clinical course after vaccination with that before vaccination (as control) in each case. All statistical analyses were performed with IBM SPSS software (IBM Corp., Armonk, NY, USA). A *P*-value of <0.05 was considered statistically significant. Because we conducted only one statistical test pertaining to our conclusion, we did not apply correction for multiple comparisons.

All experiments were independently repeated twice.

**Reporting summary.** Further information on research design is available in the Nature Research Reporting Summary linked to this article.

## Data availability

All data supporting the findings of this study are available within the article and its Supplementary Information Files and from the corresponding author on reasonable request.

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

## Acknowledgements

The authors thank Ms. Yukiko Matsushima and Mr. Kazuto Hagimura of the Keio University Hospital Clinical and Translational Research Center (CTR) for monitoring, Mr. Ryo Takemura and Mr. Ryota Ishii of the Keio University Hospital CTR for assistance with statistical analysis, and Ms. Naoko Tsuzaki of the Department of Neurosurgery for technical assistance with laboratory procedures. We also thank Ms. Akemi Hori, Ms. Emiko Seki, Ms. Kaori Kaseda, and Ms. Kana Watanabe of the Department of Otorhinolaryngology for performing the hearing examinations. We thank Dr. Owen Proudfoot of the Edanz Group (www.edanzediting.com/ac) for editing a draft of this paper. This work was supported in part by grants from the Japan Society for the Promotion of Science (JSPS) (17H04306 to M.T., 18K08951 to R.U.), and by the Japan Agency for Medical Research and Development (19lm0203088h0001 to M.T.).

## Author contributions

MT is the principal investigator of this study. RT, MF, YM, KO, KK, YO, MS, NO, HF, and SN were study investigators. RT and MT did the statistical analysis. RT, MF, RU, NO, KO, YK, TO, KY and MT designed the clinical trial. RT, MF, YM, KO, KK, YO, MS, NO, HF, TH, and SN collected the data. RT, MF, YM, KO, KK, YO, MS, NO, HF, SN, KO, YK, TO, KY, and MT analyzed and interpreted data. All authors contributed to the writing of the report and approved the final version.

## Competing interests

T.H. is an employee of OncoTherapy Science, Inc. All the other authors declare no competing interest.
