## [Peer Review File · Nature Communications]

Reviewers' comments:

Reviewer #1 (Remarks to the Author); expert in NF2 clinical trials:

I read with a great interest this paper showing an interesting new immunotherapeutic strategy for schwannomas in NF2 patients, using VEGFRs peptide vaccination. At first sight the results are interesting and very innovative and could be an alternative to Bevacizumab which works pretty well already.

This treatment seems to be safe and has been previously used in glioblastomas (same team). On the other hand, it exists improvisations and questions in the analysis of the results which require clarifications to validate them.

The NF2 scientific community has published one paper to report the hearing outcome for NF2 patients –Plotkin SR et al, Neurology 2013). Moreover, at least one publication (Blakeley et al, 2016 JCO) has described the way the results should be analyzed, for hearing and tumor volume responses.

I would get responses to the following points:

1/ page 5: NF2 patients were included because of growing VS. The authors did not show the tumor growth of each VS (or schwannoma) before inclusion and under treatment. A large figure with a curve showing the compiled evolution of each tumor before and during the treatment (after the 8 cycles completion) and at 1 year should be added. The table 1 is not precise and the figures are too small to be understood.

2/ page 6: outcome and assessments. I am surprised that the authors have used CT or MRI in 2016 to analyze the volume of the relatively small schwannomas. Why CT was used and in how many patients for how many measurements? This is a major weakness for the analysis of the results and a surprising protocol.

3/ for the tumor volume response, the authors have considered a radiological response with volumetric reduction >20% which is almost correct: it is only a PR (partial response) ? The authors should have described the tumor response using the Bevacizumab/ Blakeley paper (JCO, 2016): MR: minor response ((decrease (-19%, PD (progressive disease): increase of tumor volume >20% and SD (stable disease) for all others.

Using this clear classification, the authors should remove the term tumor decrease (less than 20%) in their manuscript because this imprecision, probably involuntary, confuses the reader. For example, page 12, they wrote: "with regard it decreased by 10% in 2 cases and increased in 2 others. They should consider: two MR, one SD, one PD. Idem for VS analysis in the same page 12.

4/ page 7, hearing outcome. The authors consider a change of at least 10% as improvement or deterioration. This definition is not accurate, not used in NF3 and in the case, makes the results better. For the subject 4, it is not hearing improvement and it is at the limit for the subject 7: the results are 1 or 2 patient improved for hearing (authors claimed 3/6, 50%!).

The authors should modify their analysis using the Plotkin Paper Neurology, 2013).

5/ I do not understand what are the 2 schwannomas that were removed and analyzed, page 12. The authors should explain this.

6/ The authors should discuss and give explanations why hearing improvement (in case 2 and 7) occurred without any tumor volume reduction.

I would like to have more details on the results of the gliomas trials done by the authors with the similar strategy. This is not developed page 15.

In conclusion, this study is innovative and potentially interesting for NF2. The authors need to improve the accuracy of their manuscript, add a large figure compiling the tumor volume variations before, under vaccination and after and answer to my questions.

Reviewer #2 (Remarks to the Author); expert in peptide vaccine for cancer:

Treatment with the anti-vascular endothelial growth factor (VEGF) antibody bevacizumab has reportedly resulted in tumor control and hearing improvement in NF2 patients, presumably because VEGF-A is essential for the growth of these tumors.

The authors demonstrated VEGFRs expression in endothelial cells and tumor cells in NF2 schwannomas, thus targeting VEGFR's is appropriate.

In this manuscript, the authors conducted a clinical study of VEGFRs peptide vaccine in 7 patients with Neurofibromatosis type II and measured the hearing improvement (50%) as well as tumor size reduction (>20%). Each patient received 8 vaccinations. However, based on only 7 patients (of which, they had different clinical outcomes) it might be a bit challenging to draw any statistical conclusions. So small sample size here is an issue.

The aim of peptide vaccine immunotherapy is to activate cytotoxic T lymphocytes (CTLs) in patients via the administration of antigens in the form of peptides.

The problems with peptide vaccination designed to elicit CTLs are well documented in the literature. NO peptide CTLs have been FDA approved. So targeting just a few CTL epitopes is unlikely to make a dent in clinical responses. There were no indication that helper T cell (CD4+) activity was included in the vaccines. Thus a robust CTL response is unlikely to occur.

Although it was mentioned that PD-L1 expression was low, check point inhibitors are required to enhance T cell responses. So that is another defect in the peptide vaccine strategy.

There are examples of other VEGFR vaccine strategy being tested , so this study is not novel.

Peptide vaccines with epitopes in the clinic are generally quite safe.

The discussion and the conclusion part of the study needs to be revised and explained more in detail. There are more papers that haven't been discussed in these sections.

We are very grateful to the reviewers for their insightful comments and suggestions, which have undoubtedly helped us to improve our manuscript immensely. As indicated in the responses below, we have taken all their comments and suggestions into account when generating the revised version of the manuscript. Responses to the reviewers' comments appear after the arrows, in blue text.

Reviewer #1:

I read with a great interest this paper showing an interesting new immunotherapeutic strategy for schwannomas in NF2 patients, using VEGFRs peptide vaccination. At first sight the results are interesting and very innovative and could be an alternative to Bevacizumab which works pretty well already.

This treatment seems be safe and has been previously used in glioblastomas (same team).

On the other hand, it exits improvisations and questions in the analysis of the results which require clarifications to validate them.

The NF2 scientific community has published one paper to report the hearing outcome for NF2 patients –Plotkin SR et al, Neurology 2013). Moreover, at least one publication (Blakeley et al, 2016 JCO) has described the way the results should be analyzed, for hearing and tumor volume responses.

→

Thank you very much for your favorable comments. In accordance with them, we have improved the analysis of the results in an effort to further validate them, and we now cite the recommended references in the revised manuscript.

I would get responses to the following points:

1. page 5: NF2 patients were included because of growing VS. The authors did not show the tumor growth of each VS (or schwannoma) before inclusion and under treatment. A large figure with a curve showing the compiled evolution of each tumor before and during the treatment (after the 8 cycles completion) and at 1 year should be added. The table 1 is not precise and the figures are too small to be understood.

→

Thank you very much for your comments. We have incorporated tumor growth pre vaccination, during the vaccination course, and post-vaccination into the manuscript, and in accordance with your comments we have now emphasized pre-vaccination and post vaccination data using larger figures with a curve, in the revised manuscript. As

you indicated, pre-vaccination imaging is important because it is known that neurofibromatosis type 2 (NF2)-related tumors sometimes exhibit prolonged stability without any intervention. Therefore, “progressive growth” has been added to the inclusion criteria in the present study. All tumors showed linear growth before vaccination, which is reflected in the relevant Figures. In addition, representative cases have been reorganized with a curve of tumor volume to facilitate an easier understanding of the effects of vaccination. A radiographic image derived from case 6 at 12 months after the first vaccination has also been included.

The new Figures 1C and 2C have been added to depict the percent change in volume from baseline (pre-vaccination), which is described in a recommended report (Blakeley, Ye, Duda, et al. 2016: <https://www.ncbi.nlm.nih.gov/pubmed/26976425>). We have also improved Table 1 to make it easier to understand the results. Table 1 has now been separated into two tables in the revised manuscript.

2. page 6: outcome and assessments. I am surprised that the authors have used CT or MRI in 2016 to analyze the volume of the relatively small schwannomas. Why CT was used and in how many patients for how many measurements? This is a major weakness for the analysis of the results and a surprising protocol.

→

In accordance with your comments we have now added to the explanation of the measurements of tumor volume, and cited recommended references. First, CT perfusion scanning was used to evaluate only tumor blood volume as a ratio of the relative values to the lesion of interest in normal-appearing white matter, as previously described (Welker K. AJNR Am J Neuroradiol. 2015; Xyda A. Eur Radiol. 2011; Beppu T. J Neurooncol. 2011). With regard to your comments and the recommended references, volumetric analysis based on MRI (cm^3) was performed to determine the complex shapes and quantify the slow growth of NF2-related tumors (Dombi E. Neurology. 2013). We also used a volume segmentation method in the analysis, because volumetric MRI analysis is known to reflect the actual size of a lesion more closely than linear measurements.

Sophisticated segmentation methods with complex algorithms have been used to define regions of interest that exhibit specific imaging characteristics (Solomon J. Comput Med Imaging Graph. 2004; Harris GJ. Neurosurgery. 2008; Weizman L. Med Biol Eng Comput 2012), and some software such as MEDx and Vitrea2 has been used for the volumetric analysis of plexiform neurofibromas and vestibular schwannomas. In our preliminary analysis we used the SYNAPSE VINCENT imaging system (Fujifilm

Medical Co., Tokyo, Japan), and the semi-automated segmentation tool function to evaluate tumor volume. However, particularly with regard to schwannomas with heterogenous intensity or with cystic components derived from V and VII/VIII nerves located adjacent to each other, the simple system involving manually outlining the target area was better than the semi-automated segmentation method using the SYNAPSE VINCENT imaging system (now shown in new supplementary Figure 4).

In the manual segmentation procedure we drew a region of interest around the lesion boundaries on each slice where the lesion was observed. The area was multiplied by the slice thickness (1 mm) to calculate the volume of the lesion within that slice, and the volumes from all slices were added to produce the total lesion volume. This method was reported to be highly accurate in previous studies (Goldmacher GV. *Br J Clin Pharmacol.* 2012; Pupulim LF. *Diagn Interv Imaging.* 2018; Veeraraghavan H. *Sci Rep.* 2018). In the present study all volumetric analysis was performed via 3.0 tesla MRI, and high-resolution post-contrast T1-weighted MRI sequences were used with the thin slice thickness of 1 mm. To improve accuracy and minimize potential effects of bias, all analyses were performed via consensus by three authors who were blind to the corresponding clinical information.

References

Dombi E, Ardern-Holmes SL, Babovic-Vuksanovic D, Barker FG, Connor S, Evans DG, Fisher MJ, Goutagny S, Harris GJ, Jaramillo D, Karajannis MA, Korf BR, Mautner V, Plotkin SR, Poussaint TY, Robertson K, Shih CS, Widemann BC; REiNS International Collaboration. Recommendations for imaging tumor response in neurofibromatosis clinical trials. *Neurology.* 2013;81(21 Suppl 1):S33-40.

Solomon J, Warren K, Dombi E, Patronas N, Widemann B. Automated detection and volume measurement of plexiform neurofibromas in neurofibromatosis 1 using magnetic resonance imaging. *Comput Med Imaging Graph.* 2004;28(5):257-65.

Harris GJ, Plotkin SR, Maccollin M, Bhat S, Urban T, Lev MH, Slattery WH. Three-dimensional volumetrics for tracking vestibular schwannoma growth in neurofibromatosis type II. *Neurosurgery.* 2008;62(6):1314-9.

Weizman L, Hoch L, Ben Bashat D, Joskowicz L, Pratt LT, Constantini S, Ben Sira L. Interactive segmentation of plexiform neurofibroma tissue: method and preliminary

performance evaluation. *Med Biol Eng Comput.* 2012;50(8):877-84.

Goldmacher GV, Conklin J. The use of tumour volumetrics to assess response to therapy in anticancer clinical trials. *Br J Clin Pharmacol.* 2012;73(6):846-54.

Pupulim LF, Ronot M, Paradis V, Chemouny S, Vilgrain V. Volumetric measurement of hepatic tumors: Accuracy of manual contouring using CT with volumetric pathology as the reference method. *Diagn Interv Imaging.* 2018;99(2):83-89.

Veeraraghavan H, Dashevsky BZ, Onishi N, Sadinski M, Morris E, Deasy JO, Sutton EJ. Appearance Constrained Semi-Automatic Segmentation from DCE-MRI is Reproducible and Feasible for Breast Cancer Radiomics: A Feasibility Study. *Sci Rep.* 2018 Mar 19;8(1):4838.

3. for the tumor volume response, the authors have considered a radiological response with volumetric reduction >20% which is almost correct: it is only a PR (partial response) ? The authors should have described the tumor response using the Bevacizumab/ Blakeley paper (JCO, 2016): MR: minor response ((decrease (-19%, PD (progressive disease): increase of tumor volume >20% and SD (stable disease) for all others. Using this clear classification, the authors should remove the term tumor decrease (less than 20%) in their manuscript because this imprecision, probably involuntary, confuses the reader. For example, page 12, they wrote: “with regard it decreased by 10% in 2 cases and increased in 2 others. They should consider: two MR, one SD, one PD. Idem for VS analysis in the same page 12.

→

In accordance with your comments we now describe tumor responses in the revised manuscript with reference to the bevacizumab/Blakeley paper (Blakeley et al. 2016, see below). Furthermore, we have removed the terms that you have correctly pointed out above are potentially problematic. Of the total of seven patients, partial responses (PR) were observed in more than 1 schwannoma in two patients (case 6 in which previous bevacizumab treatment had previously been ineffective, and case 4), and minor responses (MR) were observed in the other five patients. Out of 23 schwannomas, PR were observed in 4, MR were observed in 11, and stable disease (SD) was observed in seven. Only one cystic tumor became enlarged 12 months after the first vaccination (progressive disease [PD]).

Reference

Blakeley JO, Ye X, Duda DG, Halpin CF, Bergner AL, Muzikansky A, Merker VL, Gerstner ER, Fayad LM, Ahlawat S, Jacobs MA, Jain RK, Zalewski C, Dombi E, Widemann BC, Plotkin SR. Efficacy and Biomarker Study of Bevacizumab for Hearing Loss Resulting From Neurofibromatosis Type 2-Associated Vestibular Schwannomas. *J Clin Oncol.* 2016;34(14):1669-75.

4. page 7, hearing outcome. The authors consider a change of at least 10% as improvement or deterioration. This definition is not accurate, not used in NF3 and in the case, makes the results better. For the subject 4, it is not hearing improvement and it is at the limit for the subject 7: the results are 1 or 2 patient improved for hearing (authors claimed 3/6, 50%!).

The authors should modify their analysis using the Plotkin Paper Neurology, 2013).

→

In accordance with your indication, we have modified the analysis using the criteria described in Plotkin's paper (Plotkin et al. 2013, see below). Based on those criteria, two patients (case 5 and 6) who exhibited word recognition scores (WRSs) ≥ 90 pre-vaccination were excluded from hearing evaluation in the present study. Therefore, improvements in hearing were observed in 2 of the 5 assessable cases (40%), and no changes were observed in the other 3 cases. These results are described in Figure 1E. It may have been difficult to improve hearing in case 1 because he had been profoundly deaf in both ears for a long time (so-called "dead ears"). It is well known that long-term profound deafness leads to the reduction of cortical hearing perception, and that often this change cannot be recovered even if there is recovery of input from the auditory nerve.

We had to consider standard language differences when conducting WRS testing (Horiguti et al. *International Audiology*. 1966). In contrast, pure-tone audiograms (PTAs) are widely used with the same sound inputs throughout the world. Therefore, in our report on the present study we are inclined to retain the results obtained via PTAs. In general, global standard protocols that can be used in patients who speak different languages are required for the process of drug development. WRS criteria as they apply in patients who speak languages other than English should be carefully considered.

References

Plotkin SR, Ardern-Holmes SL, Barker FG 2nd, Blakeley JO, Evans DG, Ferner RE, Hadlock TA, Halpin C; REiNS International Collaboration. Hearing and facial function

outcomes for neurofibromatosis 2 clinical trials. *Neurology*. 2013;81(21 Suppl 1):S25-32.

Horiguti S. Comparison of speech audiometry test-words among various languages. *International Audiology*. 1966; 2:275-279.

5. I do not understand what are the 2 schwannomas that were removed and analyzed, page 12. The authors should explain this.

→

In accordance with your comment we have now added an explanation about the histological changes in schwannomas post-vaccination. Four tumor samples were paired pre vaccination/post-vaccination samples obtained from two NF2 patients. A trigeminal schwannoma was partially removed at the time of an intracranial hemorrhage 5 months after the last vaccination in case 1. In case 2, although a skin schwannoma had stabilized it was completely removed 11 months after the last vaccination at the patient's request, for solely cosmetic reasons. Both tumors exhibited insubstantial changes in volume (stable disease, rather than a positive response or progressive disease). As we state in the Adverse events sub-section of the revised manuscript, dural arteriovenous fistula may have been the origin of hemorrhaging in case 1. The Independent Data Monitoring Committee alluded to in the manuscript concluded that the experimental vaccine was not likely to have been related to this adverse event.

Paired pre-vaccination/post-vaccination tissues from the same patients were extremely valuable due to their relative rarity, and they were powerful tools for investigating the biological effects of targeted therapies. Significant reductions in VEGFR expression in schwannomas were observed in those post-vaccination tumor samples.

6. The authors should discuss and give explanations why hearing improvement (in case 2 and 7) occurred without any tumor volume reduction.

→

In accordance with your comment, we have added a possible explanation for hearing improvement without any tumor volume reduction to the Discussion section of the revised manuscript. As well as direct compression of auditory nerve fibers by tumors, in cases of NF2-associated deafness detrimental paracrine substances such as proinflammatory cytokines from tumors have been proposed as a mechanism of cochlear hearing loss (Fujioka et al. *Front Pharmacol*. 2014). During vaccination in the present study, PTA thresholds did not change in 13 of 14 ears (the threshold deteriorated

in 1 ear). In contrast, WRS either improved or remained stable. These observations imply that the ameliorating effects of VEGFR peptides may be due to a reduction of paracrine effects associated with tumors. In a previous report VEGF/VEGFR signaling was associated with proinflammatory cytokines (Waldner MJ. J Exp. Med. 2010). Further investigation is warranted to confirm the findings of the present study.

References

Fujioka M, Okano H, Ogawa K. Inflammatory and immune responses in the cochlea: potential therapeutic targets for sensorineural hearing loss. Front Pharmacol. 2014;5:287.

Waldner MJ, Wirtz S, Jefremow A, Warntjen M, Neufert C, Atreya R, Becker C, Weigmann B, Vieth M, Rose-John S, Neurath MF. VEGF receptor signaling links inflammation and tumorigenesis in colitis-associated cancer. J Exp Med. 2010;207(13):2855-68.

7. I would like to have more details on the results of the gliomas trials done by the authors with the similar strategy. This is not developed page 15.

→

In accordance with your comment we have now added more detailed information pertaining to the glioma trial that utilized a similar strategy. We previously reported a pilot clinical study investigating VEGFR1 and VEGFR2 peptide vaccination in malignant glioma patients, in which the treatment exhibited safety and yielded therapeutic effects in some patients. Eight patients received vaccinations weekly at the same dose 8 times. The first four vaccines induced positive immune responses against at least one of the targeted VEGFR epitopes in the peripheral blood mononuclear cells in 87.5% of patients. The median overall survival time in all patients was 15.9 months, which was prolonged compared with historical control data. Two patients exhibited progression-free status lasting at least 6 months. During the vaccination period, one patient developed an ulcer at the injection site (grade 3). No other adverse events were detected in that study. Based on those results, a clinical trial investigating VEGFR1 and VEGFR2 peptide vaccination is now underway in patients with primary glioblastoma.

8. In conclusion, this study is innovative and potentially interesting for NF2. The authors need to improve the accuracy of their manuscript, add a large figure compiling the tumor volume variations before, under vaccination and after and answer to my

questions.

→

Thank you very much for your favorable comments. Based on your observations and suggestions we have improved the accuracy of the manuscript, and added large figures encompassing the periods before, during, and after vaccination. In the revised manuscript we have addressed all of the points you have raised.

Reviewer #2:

1. Treatment with the anti-vascular endothelial growth factor (VEGF) antibody bevacizumab has reportedly resulted in tumor control and hearing improvement in NF2 patients, presumably because VEGF-A is essential for the growth of these tumors. The authors demonstrated VEGFRs expression in endothelial cells and tumor cells in NF2 schwannomas, thus targeting VEGFR's is appropriate.

In this manuscript, the authors conducted a clinical study of VEGFRs peptide vaccine in 7 patients with Neurofibromatosis type II and measured the hearing improvement (50%) as well as tumor size reduction (>20%). Each patient received 8 vaccinations. However, based on only 7 patients (of which, they had different clinical outcomes) it might be a bit challenging to draw any statistical conclusions. So small sample size here is an issue.

→

By way of acknowledging the point you make above, we have now added the following sentences as a “limitations” statement, to the end of the Discussion section in the revised manuscript:

The main limitation of the present study was the small sample size. The prevalence of NF2 is reportedly approximately 1 in 25,000 to 1 in 33,000 (Plotkin SR. Semin Neurol. 2018). In addition, HLA type matching restricted the number of patients enrolled.

Although studies with a larger number of NF2 patients are warranted to confirm the findings of the present study, the results suggest that this immunotherapeutic strategy may be favorable in patients with benign tumors such as NF2.

References

Plotkin SR, Wick A. Neurofibromatosis and Schwannomatosis. Semin Neurol. 2018;38:73-85.

2. The aim of peptide vaccine immunotherapy is to activate cytotoxic T lymphocytes (CTLs) in patients via the administration of antigens in the form of peptides.

The problems with peptide vaccination designed to elicit CTLs are well documented in

the literature. NO peptide CTLs have been FDA approved. So targeting just a few CTL epitopes is unlikely to make a dent in clinical responses. There were no indication that helper T cell (CD4+) activity was included in the vaccines. Thus a robust CTL response is unlikely to occur.

→

As you have correctly indicated, peptides containing only CD8-positive T cell epitopes have typically had insufficient effects on malignant tumors, and no peptide vaccines designed to elicit CTL responses have been approved by the Food and Drug Administration of the USA. Notably however, the focus of the present study was to investigate the immunotherapeutic effects of peptide-based vaccination on slow growing benign tumors such as NF2 schwannomas. To the best of our knowledge no similar study has been reported. Peptides containing only CD8-positive T cell epitopes exhibited preliminary efficacy in NF2 patients in the present study. As you have indicated, vaccines inducing CD4-positive T helper cells are important in the generation of antitumor responses. Dual activation of tumor-associated antigen-specific CTLs and T helper cells by dendritic cell vaccines have reportedly induced superior clinical responses to single CTL epitope vaccine in patients with cancers such as melanoma (Kumai T. *Cancer Immunol Res.* 2017; Kumai T. *Curr Opin Immunol.* 2017; Koido S. *Clin Cancer Res.* 2014; Muranski P. *Blood.* 2008; Tran E. *Science.* 2014; Quezada SA. *J Exp Med.* 2010; Slingluff CL. *Clin Cancer Res.* 2013). Therefore, a combination of CD8 and CD4 epitopes would be a better strategy for increasing the efficacy of peptide vaccination in NF2 patients. Based on your comments, we have now added sentences alluding to this to the Discussion section of the revised manuscript.

References

Kumai T, Lee S, Cho HI, et al. Optimization of Peptide Vaccines to Induce Robust Antitumor CD4 T-cell Responses. *Cancer Immunol Res.* 2017;5(1):72-83.

Kumai T, Kobayashi H, Harabuchi Y, et al. Peptide vaccines in cancer-old concept revisited. *Curr Opin Immunol.* 2017;45:1-7.

Koido S, Homma S, Okamoto M, Takakura K, Mori M, Yoshizaki S, et al. Treatment with chemotherapy and dendritic cells pulsed with multiple Wilms' tumor 1 (WT1)-specific MHC class I/II-restricted epitopes for pancreatic cancer. *Clin Cancer Res.* 2014;20:4228–39.

Muranski P, Boni A, Antony PA, Cassard L, Irvine KR, Kaiser A, et al. Tumor-specific Th17-polarized cells eradicate large established melanoma. *Blood*. 2008;112:362–73.

Tran E, Turcotte S, Gros A, Robbins PF, Lu YC, Dudley ME, et al. Cancer immunotherapy based on mutation-specific CD4+ T cells in a patient with epithelial cancer. *Science*. 2014;344:641–5.

Quezada SA, Simpson TR, Peggs KS, Merghoub T, Vider J, Fan X, Blasberg R, Yagita H, Muranski P, Antony PA, et al. Tumor-reactive CD4(+) T cells develop cytotoxic activity and eradicate large established melanoma after transfer into lymphopenic hosts. *J Exp Med*. 2010;207:637–650.

Slingluff CL, Jr, Lee S, Zhao F, Chianese-Bullock KA, Olson WC, Butterfield LH, Whiteside TL, Leming PD, Kirkwood JM. A randomized phase II trial of multiepitope vaccination with melanoma peptides for cytotoxic T cells and helper T cells for patients with metastatic melanoma (E1602) *Clin Cancer Res*. 2013;19:4228–4238.

3. Although it was mentioned that PD-L1 expression was slow, check point inhibitors are required to enhance T cell responses. So that is another defect in the peptide vaccine strategy.

→

In response to your comment we have now added the following sentences to the relevant part of the Discussion section:

PD-L1 suppresses antitumor immunity and promotes tumor progression because it tolerizes tumor-reactive T cells. In the present study low PD-L1 expression was observed in the schwannomas of most NF2 patients. In a previous study PD-1 or PD-L1 blockade therapy was a preferable treatment option, even in patients who were PD L1 negative (Shen X. *BMJ*. 2018). Immune checkpoint inhibition may exert synergic effects when administered with this type of CTL mediated antitumor immunotherapy in NF2 patients.

Reference

Shen X, Zhao B. Efficacy of PD-1 or PD-L1 inhibitors and PD-L1 expression status in cancer: meta-analysis. *BMJ*. 2018;362:k3529.

4. There are examples of other VEGFR vaccine strategy being tested , so this study is not

novel. Peptide vaccines with epitopes in the clinic are generally quite safe.

The discussion and the conclusion part of the study needs to be revised and explained more in detail. There are more papers that haven't been discussed in these sections.

→

As you indicate, clinical trials investigating peptide-based vaccine therapy using VEGFR derived epitopes have been conducted previously to assess safety, tolerability, and potential clinical activity in patients with advanced pancreas, gastrointestinal cancers and renal cell cancers. In addition, we have previously conducted exploratory clinical studies investigating VEGFR peptide vaccination with and without multiple glioma oncoantigens in patients with recurrent high-grade gliomas, wherein the treatment exhibited safety and yielded therapeutic effects in some patients.

We had reviewed all clinical trials using a VEGFR1 and/or VEGFR2 vaccine strategy, and in accordance with your comments we have now also generated a new table (“Supplementary Table 3”) that readers can consult in this regard (Miyazawa M. Cancer Sci. 2010; Hazama S. J Transl Med. 2014; Hazama S. J Transl Med. 2014; Iinuma H. J Transl Med. 2014; Ishizaki H. Clin Cancer Res. 2016; Masuzawa T. Int J Oncol. 2012; Wada S. Cancer Res. 2015; Yoshimura K. Br J Cancer. 2013). Concordant with similar observations in previous clinical trials, the safety of the vaccine used in the present study was evident in patients with progressive NF2 schwannomas.

As you state above, VEGFR1 and VEGFR2 peptide vaccination is not novel. However, the concept of using this immunotherapeutic strategy in patients with benign tumors is novel. We have added appropriate sentences to this effect to the Discussion section, and modified the Conclusion section accordingly.

References

Miyazawa M, Ohsawa R, Tsunoda T, Hirono S, Kawai M, Tani M, Nakamura Y, Yamaue H (2010) Phase I clinical trial using peptide vaccine for human vascular endothelial growth factor receptor 2 in combination with gemcitabine for patients with advanced pancreatic cancer. Cancer Sci 101:433-439

Hazama S, Nakamura Y, Takenouchi H, Suzuki N, Tsunedomi R, Inoue Y, Tokuhisa Y, Iizuka N, Yoshino S, Takeda K, Shinozaki H, Kamiya A, Furukawa H, Oka M (2014) A phase I study of combination vaccine treatment of five therapeutic epitope-peptides for metastatic colorectal cancer; safety, immunological response, and clinical outcome. J Transl med 12:63.

Hazama S, Nakamura Y, Tanaka H, Hirakawa K, Tahara K, Shimizu R, Ozasa H, Etoh R, Sugiura F, Okuno K, Furuya T, Nishimura T, Sakata K, Yoshimitsu K, Takenouchi H, Tsunedomi R, Inoue Y, Kanekiyo S, Shindo Y, Suzuki N, Yoshino S, Shinozaki H, Kamiya A, Furukawa H, Yamanaka T, Fujita T, Kawakami Y, Oka M (2014) A phase II study of five peptides combination with oxaliplatin-based chemotherapy as a first-line therapy for advanced colorectal cancer (FXV study). *J Transl med* 12:108

Iinuma H, Fukushima R, Inaba T, Tamura J, Inoue T, Ogawa E, Horikawa M, Ikeda Y, Matsutani N, Takeda K, Yoshida K, Tsunoda T, Ikeda T, Nakamura Y, Okinaga K (2014) Phase I clinical study of multiple epitope peptide vaccine combined with chemoradiation therapy in esophageal cancer patients. *J Transl Med* 12:84

Ishizaki H, Tsunoda T, Wada S, Yamauchi M, Shibuya M, Tahara H (2006) Inhibition of tumor growth with antiangiogenic cancer vaccine using epitope peptides derived from human vascular endothelial growth factor receptor 1. *Clin Cancer Res* 12:5841-5849

Masuzawa T, Fujiwara Y, Okada K, Nakamura A, Takiguchi S, Nakajima K, Miyata H, Yamasaki M, Kurokawa Y, Osawa R, Takeda K, Yoshida K, Tsunoda T, Nakamura Y, Mori M, Doki Y (2012) Phase I/II study of S-1 plus cisplatin combined with peptide vaccines for human vascular endothelial growth factor receptor 1 and 2 in patients with advanced gastric cancer. *Int J Oncol* 41:1297-1304

Wada S, Tsunoda T, Baba T, Primus FJ, Kuwano H, Shibuya M, Tahara H (2005) Rationale for antiangiogenic cancer therapy with vaccination using epitope peptides derived from human vascular endothelial growth factor receptor 2. *Cancer Res* 65:4939-4946

Yoshimura K, Minami T, Nozawa M, Uemura H (2013) Phase I clinical trial of human vascular endothelial growth factor receptor 1 peptide vaccines for patients with metastatic renal cell carcinoma. *Br J Cancer* 108:1260-1266

Kikuchi R, Ueda R, Saito K, Shibao S, Nagashima H, Tamura R, Morimoto Y, Sasaki H, Noji S, Kawakami Y, Yoshida K, Toda M. A Pilot Study of Vaccine Therapy with Multiple Glioma Oncoantigen/Glioma Angiogenesis-Associated Antigen Peptides for Patients with Recurrent/Progressive High-Grade Glioma. *J Clin Med*. 2019 Feb 20;8(2). pii: E263

REVIEWERS' COMMENTS:

Reviewer #1 (Remarks to the Author):

I am pleased with the responses of the authors.

The volume of the tumors is of great interest.

As mentioned, the number of patient is small but one keep in mind that Bevacizumab for NF2 was initially published in NEJM with 10 patients.

Michel Kalamarides MD PhD

Reviewer #2 (Remarks to the Author):

The authors have responded favorably to comments by reviewers.

Thus, the manuscript has been updated and can be published.

We are very grateful to the reviewers for their insightful comments. Responses to the reviewers' comments appear after the arrows, in blue text.

Reviewer #1:

I am pleased with the responses of the authors.

The volume of the tumors is of great interest.

As mentionned, the number of patient is small but one keep in mind that Bevacizumab for NF2 was intially published in NEJM with 10 patients.

Michel Kalamarides MD PhD

→

Thank you very much for your favorable comments.

Reviewer #2:

The authors have responded favorably to comments by reviewers.

Thus, the manuscript has been updated and can be published.

→

Thank you very much for your favorable comments.